# Understanding Attentional Functioning in Adult Attention Deficit Hyperactivity Disorder—Could This Improve Diagnostic Specificity?

**DOI:** 10.3390/ijerph20065077

**Published:** 2023-03-14

**Authors:** Diana Babajanyan, Leanne Freame, Ray Steele, Alison Poulton

**Affiliations:** 1Department of Psychology, Macquarie University, Macquarie Park, NSW 2109, Australia; 2Faculty of Medicine, Health and Human Sciences, Macquarie University, Macquarie Park, NSW 2109, Australia; 3ADDults with ADHD, 3/51 Wicks Rd., North Ryde, NSW 2113, Australia; 4Brain Mind Centre Nepean, University of Sydney, 62 Derby St., Kingswood, NSW 2747, Australia

**Keywords:** ADHD in adulthood, patterns of attention, attentional dysfunction, qualitative research, mental effort reward imbalances model of ADHD, reward deficiency

## Abstract

The diagnostic criteria for attention deficit hyperactivity disorder (ADHD) reflect the behavioural and functional outcomes of cognitive processes. Historically they have been based on external observations and lack specificity: clinical cohorts of children meeting diagnostic criteria show that around 40% may also meet diagnostic criteria for oppositional defiant disorder (ODD). We have proposed a clinical model to explain this: the Mental Effort Reward Imbalances model of ADHD (MERIM). This model views the lower levels of task completion that underlie several of the diagnostic criteria for ADHD as being due to a summation of deficits in executive functioning and reward processing. The subjective experience of inadequate reward from task completion may explain the reduced motivation, negativity, and oppositional attitude associated with ODD. The hypothesis for this study is that descriptions of affected individuals’ attentional characteristics could be more specific for the executive functioning deficits associated with ADHD than the current symptom-based approaches. To test whether this might be usable in practice, we conducted a workshop that aimed to characterise in depth the patterns of attention experienced by adults with ADHD and how they impact functioning. Three main patterns were described: (1) complete lapses in attention; (2) partial attention to a task; (3) attending to multiple tasks and distractions, either simultaneously or in rapid sequence. All of these resulted in reduced productivity. They also described strategies for managing their attention deficits. Some people used distractions positively, to stimulate the mind to remain active and engaged rather than losing focus. Multi-tasking could also achieve this by providing higher levels of stimulation, however, the stimulation could itself become a distraction. Interest or stress might maintain engagement; extremes could sometimes lead to hyperfocusing, which was typically infrequent but could be highly productive. Focusing on executive functions may improve diagnostic sensitivity, as the current criteria fail to identify people who function adequately due to their use of strategies that mitigate the effects of their attentional deficits. Such people may present with secondary depression or anxiety rather than clear, behavioural symptoms of ADHD. With further development, the approach described in this paper may provide a more simple and fundamental way of recognising ADHD within the community. In the longer term, focusing more specifically on executive functions may provide cohorts with a ‘purer’ form of ADHD for scientific study.

## 1. Introduction

The increased awareness of ADHD occurring in adulthood is associated with substantial diagnostic challenges. The present paper considers whether a new approach to diagnosing ADHD based on attentional characteristics might be more specific and sensitive for the executive functioning deficits in adult ADHD than the current symptom-based approaches and the particular emphasis placed on functioning during childhood. 

### 1.1. Lack of Specificity to Current Diagnostic Processes for ADHD

The behavioural characteristics associated with attention deficit hyperactivity disorder (ADHD) [1,2] or hyperkinetic disorder (HD) [3] have been recognized for many years. In 1871, Charles Darwin referred to a type of person with restlessness and lack of occupational consistency: ‘Restless men who will not follow any steady occupation—and this relic of barbarism is a great check to civilisation—emigrate to newly-settled countries, where they prove useful pioneers’ [4]. Over the years since then, more proscriptive diagnostic criteria have been proposed. The condition is currently categorised by two distinct classification systems, each with its own diagnostic criteria. These are from the Diagnostic and Statistical Manual of the American Psychological Association (DSM), which describes attention deficit hyperactivity disorder (ADHD) [1,2], and the World Health Organisation International Classification of Diseases (ICD), which describes hyperkinetic disorder (HD) [3]. Each of these diagnoses comes with several subtypes.

There is substantial overlap in the diagnostic criteria of the DSM and ICD for ADHD and HD respectively. However, they do have differences and, as a result, they identify different cohorts. In a representative sample of German children, the prevalence of ADHD according to DSM-IV was 5% compared to only 1% for ICD-10 HD [5].

DSM and ICD both have in common that they are symptom-based and describe the observable behavioural outcomes of mental processes. The main symptoms are categorized as inattention, hyperactivity/hyperkinesis, and impulsivity. Given that these lack specificity, there is the proviso that the problems cannot be better explained by another condition, which would have to be considered first. For example, a child who does not appear to listen when spoken to might be deaf, or alternatively, might have a problem understanding spoken language. One particular problem in adulthood is the frequent association of ADHD with substance use disorder (SUD), which can itself be associated with attentional problems [6].

A large cohort study of children with ADHD identified the lack of specificity for the DSM-IV diagnostic criteria, finding that 40% of diagnosed children also fulfilled the DSM-IV diagnostic criteria for oppositional defiant disorder (ODD) [7]. No explanation for this diagnostic overlap was suggested by the authors. In our own unpublished data, we have similarly found diagnosable ODD in 40% of a clinical cohort, with a further 40% having features of ODD but not reaching the diagnostic threshold; around 20% of children with ADHD appeared to have no features of ODD.

### 1.2. Impact of Current Diagnostic Processes on the Neuroscience of ADHD

The inferential lack of specificity for attentional deficits in the diagnostic criteria is likely to represent a significant problem for investigating the neuroscience of ADHD using cohorts diagnosed according to currently accepted criteria. Hoogman et al. identified differences between 1713 people with ADHD and 1529 controls in subcortical structures of the brain, including the amygdala which has a role in emotional regulation [8]. Although study selection criteria for ADHD cohorts typically involves screening for other diagnoses, these do not necessarily include ODD and its related diagnoses (e.g., conduct disorder, antisocial personality disorder) [9]. The underlying features of these conditions may well have confounded the neuroscientific study of ADHD. Similarly, although people with ADHD may perform poorly at tests of vigilance and sustained attention, their particular deficits show individual variability, which means that objective tests designed to assist with diagnosis often lack sensitivity [10].

### 1.3. ADHD as a Condition with Deficits in Executive Functioning and Reward Processing: The Mental Effort Reward Imbalances Model of ADHD (MERIM) [11,12]

We felt that a 40% overlap of ODD with ADHD was too great to be a chance association. We hypothesized that the DSM-IV diagnostic criteria were not specific for ADHD and were systematically also selecting children with ODD. We proposed a model to explain this association, which we called the Mental Effort Reward Imbalances model of ADHD (MERIM) [11,12] (Figure 1). This is an intuitive model, based on many years of experience working with people with ADHD and their families. It draws its validity from its value and acceptability as an explanation in a treatment setting.

Several of the DSM diagnostic criteria relate to task completion, for example, reluctance to start or to persist with tasks requiring sustained mental effort. The result of this is less achievement. However, although attentional deficits could result in less achievement due to the greater level of effort or determination required, this is not the only possible mechanism. We suggested that ODD could be considered a reward deficiency syndrome [13], such that task completion would result in an inadequate level of satisfaction and therefore reduce the level of motivation. Less satisfaction could result in a lower mood, higher levels of anger and irritability, and maladaptive reward-seeking behaviour, which might explain much of the symptomatology of ODD. We proposed that these two independent mechanisms would have a summative effect on an individual meeting the diagnostic criteria for ADHD (Figure 1).

Although the MERIM was derived in a clinical rather than a strictly scientific research setting, there is scientific evidence for reward processing deficits [14] and a number of studies cohort studies of children and adults with ADHD have identified problems with motivation and reward. Activation of the striatal structures of the brain, including the nucleus accumbens, occurs during the anticipation of reward [9]. Hoogman and others have observed a lower level of task-related striatal activation among adults with ADHD [9]. Sonuga-Barke noted cognitive heterogeneity and proposed two distinct pathways underlying a diagnosis of ADHD: one involving executive functioning and the other involving reward and motivational processes [15].

### 1.4. Inadequacy of Observational Symptom-Based Diagnosis of ADHD in Adults

Although ADHD was initially considered to be a condition of childhood, it has become apparent that it may affect up to 5% of adults [16]. The diagnostic criteria for children are based on behaviour that a parent or teacher might observe in a child. The diagnostic criteria for adults are modelled on the childhood criteria and have not been fully adapted for developmental age [17]. The particulars have been only slightly changed, for example losing ‘keys or mobile phone’ rather than ‘books or pencils’, and the required number of diagnostic criteria reduced by one. However, they could still be framed as: ‘Would an observer notice that you often lose your keys or mobile phone?’ For adults with a mature level of insight, assessment for ADHD should involve greater depth of the characterisation of the lived experiences of the individual.

Current diagnostic processes for diagnosing ADHD in adults often rely on retrospective evidence of attentional deficits during childhood, which may be difficult to substantiate and subject to recall bias [18,19]. Another significant problem with symptom-based diagnosis is that in adults, symptoms of ADHD may be masked by well-developed coping mechanisms [20]. For example, a person may never lose their keys because they have learned to put them away as soon as they get home. Such coping mechanisms likely contribute to the underdiagnosis of ADHD in adults and necessitate a more detailed and specific assessment of the impairment [20].

Adults who have grown up with undiagnosed ADHD may have developed comorbidities such as depression or anxiety. These may mask the underlying ADHD, with difficulties in concentration and motivation being attributed to a secondary disorder. Co-existing substance use disorder can also complicate the diagnosis of ADHD [6,19].

The current symptom-based criteria not only lack specificity but also overlook the likelihood that adults could supply a more detailed and insightful and perhaps a more specific description of their attentional deficits.

In this study, we aimed to look beyond a symptom-based approach to ADHD and to examine in greater depth the individuals’ perceptions of the cognitive processes underlying their ADHD-related functional impairment. This may also lead to a greater understanding of the ways that attention has effects on daily functioning.

### 1.5. Hypothesis

Detailed exploration of the attentional processes of adults with ADHD may increase the sensitivity of the diagnosis. This approach may also provide greater specificity by allowing the attentional deficits to be differentiated from reward deficiencies.

However, it is likely that this methodology will identify a slightly different cohort from those diagnosed using the existing diagnostic criteria. 

In the longer term, if such detailed inquiry turns out to be more specific for the executive functioning deficits associated with ADHD than the current diagnostic criteria, this could provide cohorts with a ‘purer’ form of ADHD for scientific study.

### 1.6. Aims and Rationale

#### 1.6.1. Short Term Aims

To investigate the experiences of adults with ADHD in relation to their attentionTo explore illustrative models or analogies for conceptualising ADHD, with a view to helping other people to have a greater understanding of the condition.

#### 1.6.2. Long Term Aims

To develop a greater understanding of the types of attentional deficits and their impact on functioning.To develop ways of conceptualizing ADHD that may help others within the community gain insight into what it means to have ADHD.

## 2. Materials and Methods

This study principally used a qualitative design in the context of a two-hour workshop conducted in 2019 by AP and RS with assistance from DB and LF. Demographic and medical data were also collected using non-identifiable questionnaires. 

Essentialist and inductive thematic analysis, modelled by Braun and Clarke [21] and Hahn [22], was conducted on the entire qualitative data set.

### 2.1. Participant Characteristics 

The 29 participants were adults recruited through an advertisement from an ADHD community support organisation. Twenty-four participants returned anonymous questionnaires: 11 women (mean age 39, range 20–65 years), 12 men (mean age 47, range 29–71 years), and one identified as ‘other’. Their mean age at diagnosis of ADHD was 32 years, a maximum of 49 years. Only five were diagnosed before the age of 18 and three did not specify any age of diagnosis. The mean estimated age that ADHD first affected their functioning was 9 years, with only two unable to recall having significant problems in childhood. Fourteen were currently taking a stimulant.

### 2.2. Researcher Characteristics

The research team came from a range of professional backgrounds and included one doctor (AP: paediatrician specializing in ADHD), one medical student (LF), one psychology student (DB), and one community representative of adults with ADHD (RS).

### 2.3. Data Analysis

The thematic analysis of the verbatim transcriptions of the meeting and email responses focused on addressing the exploratory research questions: “what are the characteristic experiences of attention of adults with ADHD”, and “what models best conceptualise these experiences”. These questions were presented in a simplified form to the participants (Appendix A). Thematic analysis followed four main stages, namely, initial coding (aka open coding), focused coding and category development, axial and thematic coding, and theory development. This was achieved through manual coding and data saturation techniques. Specifically, the researchers undertook an iterative process of reading and labelling the data using codes, sorting codes into subthemes, and organising subthemes into overarching categories, until no new themes were observed. To ensure that themes were internally coherent and a reliable representation of the data, a coding sheet was used to label data, and themes were clearly defined.

## 3. Results

A total of ten overarching categories emerged from the thematic analysis, three of which were dropped as they were not directly relevant to the study’s research questions. Data saturation was achieved for the retained categories for the research question: “what are the characteristic experiences of attention of adults with ADHD”. However, the models to conceptualise ADHD were diverse and although they could be sorted into themes, which showed some consistency with the descriptions of attention, it is likely that data saturation was not reached. The thematic analysis is shown in Appendix A.

All participants mentioned at least one way in which ADHD impaired their daily functioning or brought them distress. Participants most frequently reported that their ADHD symptoms affected their studies and work; some also reported impairments in their social life. Specific problems included not getting things completed and missing deadlines, mental fatigue, exhaustion, anger or frustration, and underachievement.

Participants agreed that completing a task required “disproportionate energy”, with one participant suggesting that he was functioning at only 40% of his intellectual capacity. When giving descriptions of their attention, participants often linked these to strategies that they used to compensate for the attentional difficulties. Some strategies needed to be carefully balanced or they could take over and themselves become distractions.

Many of the participants stated that their ADHD was only recognised in adulthood. For some, this was a recent realisation after ADHD was diagnosed in their child. Their awareness of their ADHD gave them new insight into the cause of their many day-to-day difficulties.

### 3.1. Specific Issues with Attention

Participants described three distinct patterns of attentional deficits: complete lapses, doing a task without fully attending, and attending only briefly due to distractions. Hyperfocusing was also described.

#### 3.1.1. Lapses in Attention

Some participants described complete lapses in attention, such that they would be unable to recall what they had carried out.


*My distractibility is so out there it’s not true, erm, to the extent I sometimes don’t even create memories of where I put something down, you know, I find things that I’ve lost.*


Two described contemporaneous difficulties with memory while waiting their turn to speak.


*Uh so actually I am going to refer to my phone because during the time that we have had this opportunity, I have forgotten at least twice what I wanted to say.*



*G’day, um I was almost going to sit down because I already forgot about what I was actually going to say um but uh now I have blanked again. What was it I was going to say?*


One participant described his perception of not attending to anything.


*I don’t really know what happens to my attention most of the time… I am not really paying attention to anything, like I am doing different things at different times, like I burn my toast all the time and that sort of thing because the attention is going all different places and never really settled on one thing, um, so what happens to it is it’s not there… It would just fly away like every other thought does.*


#### 3.1.2. Partial Attention

Participants described doing a task without focusing on it. This was particularly common for reading tasks and was often described as not processing the information.


*I can sit there, read the chapters word for word, over and over again, and not process a thing… I don’t remember, I don’t understand what I have just read. And then I get home and be like ‘oh I am so exhausted from a day of studying nothing’.*


#### 3.1.3. Attention Flitting from One Task to Another

Several participants commented that they were unable to work on a task efficiently because they would get distracted by other tasks and would forget what they had planned to do first.


*My attention sort of goes left, right, and centre and jumps all over the place and I start one task and then I think “Oh I have got to do, oh look, I’d rather do this one now”. So, I start doing that task and then the next thing pops into my head and I start doing the third task and then I’ll realise “Oh I should have actually stuck with the first task”.*



*With writing an email um I’ve got to look at what the previous email was, make sure that I answer to those points, and then also say what else I want to say but by the time I remember all those things to do, I have already checked the next email and I have forgotten everything from the first one.*



*Your brain’s … giving you too many tasks that when you’re just trying to do one task in hand, whereas most people have that filtration system that can go okay this is task A, sit down and do it, alright now I’ve done task A where’s task B.*


#### 3.1.4. Hyperfocusing

Several participants described the ability to ‘hyperfocus’. This was typically infrequent and could be positively associated with exhaustion. One man explained that at a stage of extreme tiredness, the constant stream of distracting thoughts might sometimes abate, leaving him able to work without distractions.


*And that comes once in a month maybe…. the half-sleepy state early morning at 3 a.m. or late nights at 2 a.m. have been really productive.*



*Like it’s due by midnight, 11:59. … then like suddenly two hours before it is due it starts just flowing crap and it’s just like “oh now I’m going to find that space”.*



*Right well once I am in the zone I could keep going for hours, but it takes hours to get there. … I would start seriously writing the essay at about 2 or 3 a.m.*



*Um, I am okay to just go ahead and work for just like 12 h on end. Like I can work for 12 h on end and that’s it, but then some days I won’t do anything at all.*


#### 3.1.5. Interest or Stimulation

Participants described an ongoing need for stimulation. Interesting or stimulating tasks could be completed; less engaging tasks could be made more interesting by adding complexity; multitasking could stimulate the mind and keep it working on a less interesting task.

#### 3.1.6. Better Concentration for Interesting/Stimulating Tasks


*I think having ADHD your mind’s constantly striving for stimulation so trying to do a task if it somewhat becomes boring your brain thinks—like—it’s almost trying to compensate.*



*You’ve got to be doing something. My mind’s got to be intellectually working. I can’t just—it’s like I can’t just switch off. You know—I’ve got to be intellectually working all the time.*


Concentration could only be sustained if a task was sufficiently interesting.


*If it’s not a puzzle I won’t do it. And I will just sit there and stare at it. And my mind will just jump off to other things.*



*So when it’s something as interesting, I can focus for up to hours, and the problem is, time just vanishes. So, it happens to me every day now ‘cause I’m enjoying what I do. So if I haven’t enjoyed what I do, then I just struggle.*


#### 3.1.7. Strategies for Increasing the Level of Stimulation

Tasks could be made more engaging by over-complicating them or attending to multiple stimuli simultaneously as a means of maintaining attention. However, such stimulation might itself become a distraction. Making simple tasks more complex and engaging could lead to mental fatigue.


*My mind … is not focusing on what I really need to get done … my mind’s kind of creating, making it more interesting but overcomplicating it, over analysing it.*



*When we sit down and watch a movie … the movie is fine and I am not bored by the movie but it is just not stimulating enough for me … Whether it’s like just messaging a friend on Facebook, or just walking around the room, or going through my drawers, or… cleaning up the room, … something else needs to be going on at the same time or else I’ll just fall asleep.*



*If I’ve got to go and clean that room or paint that room … the radio has to be on all the time, erm, because I think it occupies and stops me some of the time being diverted or distracted.*



*So, if you’ve got a really dull task like housework or tidying the house I will go from one end, I’ll do it in an interesting way … if I am listening to a podcast I could spend you know a couple of hours. I could spend ages if … there’s not something so exciting that it’s pulling my attention.*


One participant found that social engagement was helpful for maintaining attention.


*My attention immediately gets better when I am talking to someone; I am more engaged in that context than when left to my own devices.*


#### 3.1.8. Stress as a Form of Stimulation

Stress and frustration were often experienced and could enhance efficiency. A few participants mentioned using stress as a form of stimulation, finding that it increased attention span. Some were able to work more productively because of elevated stress and “panic” associated with an approaching deadline.


*If that human person had been stimulated, they’d focus on a task much longer than if a person wasn’t. So, as a child, I would use stress as a form of stimulation.*



*If I was stressed … my concentration would last 3 h, for the entire length of the exam. If I wasn’t under that environment it would last 5 min.*



*I find if I’ve got tasks which are really pressure I can concentrate…*


One participant found that even the increased anxiety associated with an impending deadline might be insufficient to stimulate their brain to be able to focus.


*So, it is 11:15 p.m., there are 44 min until the deadline and my attention has very quickly been distracted somewhere else. This may be in combination with the anxiety that it’s approaching. This may be in combination with my brain being completely in a zombie state or whatever reason it’s like I am not recognising that that deadline is in 45 min time so I’m often in that sort of numb state. And I would say that my anxiety increases even when my brain is still dead and I am not focused.*


#### 3.1.9. Procrastination

Participants described difficulty getting started, often due to postponing the task or getting distracted and doing something else. Sometimes seeking out distraction appeared to be a purposeful effort to postpone the task. Participants recognized their aversion to tasks that should have been manageable.


*And then, er, if it’s a dull task it literally can feel painful to try to do it so, erm, even if I know it’s good for me*



*When it’s up to me, I can’t focus, I don’t know where to start. I know that I’ve got to get rid of this stuff, but it’s a struggle to get started.*



*I have been told that if I don’t like something I don’t do it… I mean a herd of wild elephants could be behind me, but I would not move. … And the thing is, erm sometimes, even at home, and I have to do a task, maybe self-care, maybe scheduling, maybe planning, these are tasks I dread, I fear. So, I don’t want to do them. So, I mean what I would rather do is have a tab of Netflix open so I mean, YouTube open on another tab, Reddit on another tab. … So, it’s just 10 different tabs.*



*The brain is going all the time. Over analysing, overthinking, etc. … huge amount of overwhelming procrastination and I found that I am going to do a task I almost open myself up to being distracted. So okay and everything is on the table ready to go … oh got to go to the loo. Something needs to be done along the way and I have completely forgotten what I was doing. Then I think, “Oh no, I am supposed to be in the toilet and then I am supposed to be doing what I am doing.” I can get distracted a dozen times within an hour and still not get to sit down and start that task and it is often when the task is somewhat overwhelming and yet there’s no good reason for it to be a problem. It’s often a thing that I am well and truly able to do, but to sustain the attention to get started…*



*I often feel like there’s an internal self-sabotaging mechanism going on ‘cause even sometimes when I have eliminated everything on the list I got no other excuses no reason to procrastinate, I’ll sit in front of the computer and I’ll switch to ABC news. And so, it’s that there’s—even at that point—there’s still a sabotaging thing happening in my head.*


### 3.2. Illustrative Models of ADHD

The participants described a variety of analogies to illustrate their experience of ADHD and specifically the way that they felt that their experience was different from that of other people.

#### 3.2.1. Intermittent Attention


*… one of those handheld whizzes—you know—chop up like veggies or whatever and its either on and fully on and its whirring … and going crazy or its totally off.*



*It’s … like a record or a track DVD or something that keeps skipping, just gets sorted and then something grabs the attention, … or somebody got their finger on the on/off button. So, you’re on, you’re off, you’re on, you’re off…*


#### 3.2.2. Distractions Grabbing the Attention


*To me, it’s basically a multi-core processor trying to do 30 tasks and I keep jumping from task to task.*



*My brain feels like a gigantic flywheel brimming with ideas and spinning at colossal speeds … but by the time I get around to engaging with the smaller wheel (society), the conversation has already shifted.*



*having 15 televisions all with 15 different television shows on them. And it would be deciding which one of those I am going to watch if not all of them all at once.*



*It’s like a million different radio channels not quite locking in.*



*Like those websites … that have all those big bright rainbow flashing things … and as you’re trying to go through the task there are all these other pop-ups that like keep coming up that kind of drag your attention.*



*having a song repeating in my head all the time or having something in my mind all the time in the background.*



*being in a room with a lot of people talking … its like being on the side of a busy highway with all the traffic.*


#### 3.2.3. Meandering Attention


*lots of little ‘rivulets’ of streams … some grains will fall away and all the water will flow down in one side … it goes off there and then it might come back.*



*a herd of bullock and you want them to plough the field. But … one keeps running off here and one keeps running off there … Or I’ll chase after one of the bullocks and then the others are just standing there.*


#### 3.2.4. Hyperfocus


*A dog … it’s doing the- a lot of things at the same time, gets easily distracted, jumping at noises. However, if you give the dog a scent, a bone to focus on, or if you throw a stick, it will go and it will work until it reaches it and brings it back to you.*


#### 3.2.5. Inability to Retain or Use Information Efficiently


*A computer without enough RAM, like it has 4 GB of RAM trying to run windows 10, it will boot in and it can run but as soon as … it tries to do tasks it—BSOD (blue screen of death) and restarts itself in recovery mode and deletes the previous task it was trying to run.*



*It’s like my hard drive needs fragmenting … all the information is there but maybe you don’t have the RAM or the consolidation or instructional information to be able to access it quickly so your hard drive really needs defragmenting.*


#### 3.2.6. Effect of ADHD Medication

Some participants incorporated the normalising effect of ADHD medication into their models.


*Imagine you have a cloud above your head and imagine there are tasks you need to do like, washing clothes, work out what to eat this week, and so on … in the ADHD mind you triple that by 10 … It all looks a mess and there is little way to sort it without medication.*



*When I take my medication it’s like adding an extra 4 GB of RAM and it operates at normal levels. The problem with this is when the meds wear off I’m back to the original 4 GB, trying to cope with the fallout and energy usage of the 8 GB, so it needs like an hour to recover.*


## 4. Discussion

The participants gave a wealth of information. Their lived experience mapped onto well-recognized associations of ADHD and in the descriptions of their day-to-day functioning, there was a level of consistency in the types of problems experienced. Participants described three main patterns of attentional dysfunction contributing to these problems. These consisted either of complete lapses in attention which could extend to losing track of time; partial attention—attempting a task but without fully concentrating; or attending to multiple tasks and distractions, either simultaneously or in rapid sequence. These resulted in not getting tasks completed. Such disorganisation could be compounded by memory difficulties: forgetting which tasks had been started and needed finishing. Interest or stress might maintain engagement; extremes could sometimes lead to hyperfocusing. This was typically an infrequent occurrence and might occur late at night, after several hours of attempted task initiation, particularly with the added pressure of an approaching deadline. Those whose attention might wane often used distractions positively, to stimulate the mind to remain active and engaged rather than losing focus. Multi-tasking could also achieve this by providing higher levels of stimulation. Some people required multiple sources of stimulation to defray boredom. One problem experienced was that the stimulation could itself become a distraction. Procrastination was common, particularly if a task was perceived as difficult or arduous. Procrastination might take the form of looking for other tasks or distractions as an excuse for postponement or simply being overwhelmed by the thought of the task.

Participants commonly reported difficulty processing and almost immediately forgetting information, which suggests that their comorbid memory difficulties may be a direct consequence of their attentional impairments. Atkinson and Shiffrin propose that in order for information to be stored and later retained as memories, the sensory stimuli must first be interpreted and encoded, which requires sufficient attentional control [23]. This may explain the difficulty adults with ADHD experience in retrieving information, as the sensory information has not been properly encoded in the first place. Previously, the literature has found that memory encoding processes are attention dependent [24] and differ at a neural level between adults with and without ADHD [25]. Baddeley’s working memory model suggests that verbal and visual information must be continuously rehearsed in order to be temporarily stored within the working memory system [26]. In line with this model, losing focus due to distracting thoughts and external stimuli disrupts the memory rehearsal processes and may lead to poor short-term retention of information and difficulty performing tasks requiring moderate working memory capacity. Impaired rehearsal processes and temporary memory storage have been previously reported in children with ADHD [27,28]; future research should examine this within the adult population.

The models of ADHD, although diverse in topic, broadly mapped onto the attentional difficulties described. Dominant themes which were prevalent in the models were the overwhelming and exhausting nature of living with ADHD and difficulty coping with attentional deficits. Whilst participants described a variety of different models, all detailing different areas of their experience, the most frequently mentioned and agreed upon model was the computer analogy. A similar model, known as the information processing model [23], has often been used by cognitive psychologists to depict and understand human cognitive processes, memory, and attention, by comparing them to equivalent computer processes or counterparts.

We observed that when asked to come up with a model describing ADHD, participants responded quite quickly and easily. This suggests a level of awareness and insight into their experience. Some participants incorporated their experience of medication for the treatment of ADHD into their model. The description of the positive benefit enriched the models and may help others to understand what it means to have ADHD treated effectively with medication. The most popular subject matter was to compare the mind with an electronic device, such as a computer. However, as developing an analogy is likely to be a creative process, applying the qualitative research methodological concept of data saturation may not be appropriate.

A strength of this study is the number of participants who contributed. They were able to listen to the responses of those who spoke to the assembled group and further develop some of the themes. Those who were unable or reluctant to address the meeting had the opportunity to record their contributions one-to-one with a member of the research team during the break, or to send them in an email. A further strength is the diversity of the research team deriving from two different disciplines: medicine and psychology; and also, the community advocate for ADHD.

A limitation is that our sample of adults with ADHD may not be representative. People who choose to attend such a meeting may be biased towards those who are more intellectually able; such a perception of themselves could be inferred in some of their responses. Some of their descriptions of living with ADHD suggested a profound impact on their day-to-day functioning. It is therefore possible that these might represent more extreme manifestations of ADHD. Of those returning questionnaires, a substantial proportion had been diagnosed relatively recently, which may suggest an underrepresentation of people whose ADHD extended from childhood. Alternatively, it could relate to a substantial pool of undiagnosed adults.

## 5. Conclusions

The findings from this qualitative study confirmed that adults with ADHD may give insightful descriptions that could characterize the experience of living with ADHD. Developing new illustrative analogies for the functional difficulties associated with ADHD may help society to become more supportive towards individuals with this disabling condition.

The participants’ perceptions of having a shorter attention span could be objectively investigated, together with the amount of time spent daydreaming. This line of research may demonstrate measurable differences between people with and without ADHD.

Detailed investigation of the attentional processes may improve diagnostic sensitivity for ADHD, as the current criteria could fail to identify people who function adequately due to their use of strategies that mitigate the effects of their attentional deficits [20]. Such people may present with depression or anxiety rather than clear symptoms of ADHD.

The inferential lack of specificity for attentional deficits in the diagnostic criteria is likely to represent a significant problem for investigating the neuroscience of ADHD using cohorts diagnosed according to currently accepted criteria. However, changing the diagnostic approach is likely to identify a slightly different cohort from the current diagnostic procedures. In the longer term, justification for this new approach may come from neuroscience, when the findings from cohorts of people with traditionally diagnosed ADHD with and without symptoms indicative of reward deficiencies can be compared.

## Figures and Tables

**Figure 1 ijerph-20-05077-f001:**
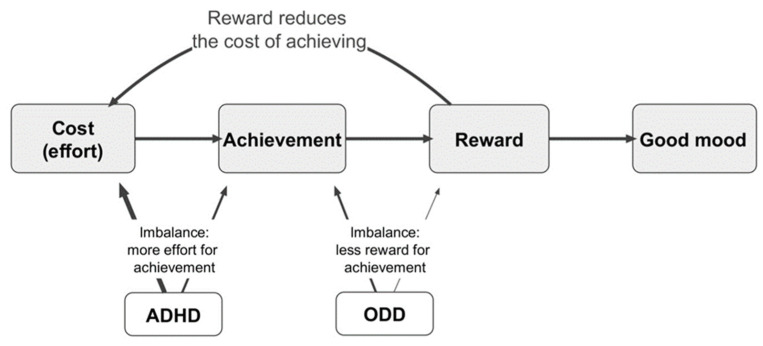
Mental Effort-Reward Imbalances Model (MERIM). Figure reproduced with permission by Australasian Psychiatry [11].

## Data Availability

The data presented in this study are available on request from the corresponding author. The data are not publicly available due to ethical and privacy reasons.

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
