# Peer review of "Understanding Attentional Functioning in Adult Attention Deficit Hyperactivity Disorder—Could This Improve Diagnostic Specificity?"

_ijerph, 2023, doi:10.3390/ijerph20065077_

Round 1
Reviewer 1 Report
I find this paper very interesting, contributing to the field of understanding the functioning of adults having ADHD. In particular 3 main patterns of “difficulties” in attention were described and all of them result in reduced productivity. I do have some suggestions that may further improve this manuscript.
I think that the Mental Effort-Reward Imbalance Model facilitates the reading of the manuscript. To my understanding, the focus of this manuscript is to investigate the “imbalance for ADHD” pathways, disregarding the ODD pathways. I think that the manus would be improved if this was shown already in the model. This is however stated in the Aims and rationale section.
The included patients, 14 of them were taking stimulants. I assume that this means that they were currently under ADHD medication, and the result are presented in the section 3.2.6. Effect of ADHD Medication. Do you think that the results differ between patients taking medication and patients not taking medication? My interpretation of the results presented in the above section is that the patients benefit in attention by their medication. Our thoughts on this may be added to the discussion.
Lastly, the results on patients describing a ADHD model. In the discussion out state that the participants easily came up with a model describing ADHD, and that this suggests a level of awareness and insight. But in the analysis section you state “However, the models to conceptualise ADHD were diverse and although they could be sorted into themes, which showed some consistency with the descriptions of attention, it is likely that data saturation was not reached”. Did you in these analysis find any common points (repeated statements) in the different models suggested? Are you able to say which are the commonest “things” included in these models, that the patients suggested?
Author Response
Thankyou for the feedback on our manuscript. I hope we have responded correctly and to the satisfaction of the reviewers.
We noticed that Table 1 was not included in the paper and have added it and Table 2 with the details of the thematic analysis.
Reviewer 1
I think that the Mental Effort-Reward Imbalance Model facilitates the reading of the manuscript. To my understanding, the focus of this manuscript is to investigate the “imbalance for ADHD” pathways, disregarding the ODD pathways. I think that the manus would be improved if this was shown already in the model. This is however stated in the Aims and rationale section.
Response – I am not sure if the reviewer is suggesting that the Mental Effort-Reward Imbalances Model should be redrawn to reflect the findings of this study. However, the model has already been published. I can give a version that shows only the ADHD imbalance, but this would be a different model.
The included patients, 14 of them were taking stimulants. I assume that this means that they were currently under ADHD medication, and the result are presented in the section 3.2.6. Effect of ADHD Medication. Do you think that the results differ between patients taking medication and patients not taking medication? My interpretation of the results presented in the above section is that the patients benefit in attention by their medication. Our thoughts on this may be added to the discussion.
Response – We did not specifically investigate the participants’ perceptions of their medication, but this did come up in some of their contributions. It is certainly possible that their experience of the difference in their functioning on medication affected their perceptions and understanding of their ADHD. We have included this in the discussion: Some participants incorporated their experience of medication for treatment of ADHD into their model. The description of the positive benefit enriched the models and may help others to understand what it means to have ADHD treated effectively with medication.
Lastly, the results on patients describing a ADHD model. In the discussion out state that the participants easily came up with a model describing ADHD, and that this suggests a level of awareness and insight. But in the analysis section you state “However, the models to conceptualise ADHD were diverse and although they could be sorted into themes, which showed some consistency with the descriptions of attention, it is likely that data saturation was not reached”. Did you in these analysis find any common points (repeated statements) in the different models suggested? Are you able to say which are the commonest “things” included in these models, that the patients suggested?
Response – In retrospect data saturation in relation to the models is perhaps not realistic as the participants’ ways of illustrating their ADHD could be considered a creative process, which would mean that saturation as such is not meaningful. We have done some classification of the models by themes, as is presented in the results. We have added the following point: The most popular subject matter was to compare the mind with an electronic device, such as a computer. However, as developing an analogy is likely to be a creative process, applying the qualitative research methodological concept of data saturation may not be appropriate.

Reviewer 2 Report
The research seems very valuable and timly one. However, since it is a qualitative study the presentation of your results should be done in a comprehensive manner. Though the thoritical parts presented in a comprehensive manner rest of the parts have not articulated in a comprehensive manner. Please refer to the detailed feedback in the manuscript.

Author Response
- Please check for reference style
Response - Our understanding is that the attached version which we have edited has been checked for reference style.
- Please remove our part here. The explanations should be objective in research manuscripts
Response - We have taken out the personal pronouns.
- Objectives should be integrated! (short term and long term objectives)
Response - The objectives have been re-written as requested
- There is very few relations of addition of rationale with the aim
Response - I am not sure what is required here. Could the reviewer please clarify what changes we should make?
- Coding mechanism can be summarised in a table
Response - We have added Table 2 with the coding
- Narrations are there. But the coding mechanism is not clearly indicated!
Response - Please refer to Table 2.
- Data could have been presented in a table where the initial coding and summary of the participants viewpoints. That could be a great understanding for the audience.
Response - The table with the coding has been added. However, we feel that the participants’ viewpoints are more easily read when integrated into the text.
- Above results have not been discussed comprehensively. Please try to have compressed them up with the coding mechanism and previous findings.
Response - Could the reviewer please give examples of discussion points we could have included?
